# Semantic Patterns of Prohibited AI Systems in the EU AI Act

Delaram Golpayegani[1,*], Harshvardhan J. Pandit[1,2] and Dave Lewis[1]

[1]*ADAPT Centre, Trinity College Dublin*
[2]*AI Accountability Lab, Trinity College Dublin*

### Abstract

The EU AI Act is a landmark piece of legislation that governs deployment and use of AI systems. Within its risk-based regime of regulation, prohibited AI practices face the strictest requirements, being entirely banned to be deployed or used within the Union. The provisions for prohibited systems have been applied since 2 February 2025. While authoritative guidelines have been published for prohibited systems, there is still no systematic approach that facilitates determination of such systems in a simplified and automated manner. To fill this gap, we specify the prohibited AI conditions, articulated in Art. 5, using combination of a minimal set of semantic concepts. We further show how these conditions can be described in a machine-readable format using semantic constraint and rule languages, such as SHACL and N3. This approach to representing prohibited rules supports a more open, interoperable, and transparent implementation of the AI Act, while also enabling partial automation of enforcement processes.

### Keywords

EU AI Act, prohibited AI, semantic rules, SHACL, N3

## 1. Introduction

The EU AI Act [1] is the first in the world AI regulation that entered into force on 1 August 2024. Adopting a risk-based approach, the AI Act regulates AI systems according to their potential risks to health, safety, and fundamental rights. Within this risk-based classification, the Act explicitly identifies two categories of AI systems: (1) *prohibited AI practices*, defined in Art. 5, and (2) *high-risk AI system*, specified in Art. 6. In addition, the Act implies another class of AI systems that impose *transparency* risks in Art. 50. Finally, it refers to *"AI systems other than high-risk"* (Art. 95), which are subject to voluntary compliance with legal obligations and are interpreted as " minimal risk AI systems". Within this categorisation, prohibited AI practices face the draconian measure of being entirely banned to be deployed or used within the Union. In case of non-compliance, providers and deployers of such systems face fines up to EUR 35 million or 7 percent of the offender's total worldwide annual turnover, whichever is higher (Art. 99(3)).

The AI Act outlines eight main categories of prohibited practices in Art. 5. Four of these categories are banned unconditionally, while the remaining four are subject to exceptions. Although the number of conditions is limited, the legal language used to describe them is vague and open to interpretation (see the discussions in [2, 3, 4]). To assist with implementation of the Act and as per Art. 96(1b), the Commission published a guideline on prohibited AI practices [5] in February 2025. While this guideline is a helpful resource to resolve ambiguities, it does not necessarily simplify the critical decision of whether an AI system is prohibited or not.

Unlike the power of adopting delegated acts for updating Annex III high-risk AI systems (Art. 7), there is no agile mechanisms to amend the list of prohibited AI systems as the AI technology as well as social preferences change. Therefore, any changes to the prohibited conditions requires following the ordinary legislative procedure, which can take several years [6]. Although the frequent changes to prohibited

*NeXt-generation Data Governance workshop 2025 (NXDG 2025), co-located with SEMANTiCS'25: International Conference on Semantic Systems, September 3–5, 2025, Vienna, Austria*

*Corresponding author.

✉ golpayes@tcd.ie (D. Golpayegani); me@harshp.com (H. J. Pandit); delewis@tcd.ie (D. Lewis)

🆔 0000-0002-1208-186X (D. Golpayegani); 0000-0002-5068-3714 (H. J. Pandit); 0000-0002-3503-4644 (D. Lewis)

systems might be unlikely, the rapid pace of changes in AI systems requires adaptable approaches that enable ongoing assessment the system's risk level to avoid any non-compliance. Motivated by the EU's initiatives for regulatory simplification [7], in this paper we aim to facilitate identification of prohibited AI systems by determining the *minimal* set of concepts that enable specifying prohibited AI systems in a way that they can be sufficiently distinguished. After conceptualisation of prohibited conditions, we demonstrate how these can be translated into codified machine-readable rules using Semantic Web technologies, particularly the Shapes Constraint Language (SHACL) [8] and Notation 3 (N3) [9]. By leveraging Semantic Web technologies, we develop a standards-based, transparent, and interoperable framework for determining prohibited AI conditions, and thereby supporting automation of compliance-related tasks. As will be discussed later, this work builds upon our previous research on determining high-risk AI systems [10], which has gained considerable traction within the community.

## 2. Related Work

Existing studies on the AI Act's prohibited AI practices (Art. 5) are primarily focused on interpreting the prohibited conditions. Some notable analyses were published prior to the publication of the AI Act in official journal of the EU, including Neuwirth's analysis of prohibited categories stated in the Commission's proposal [11], Bermúdez et al.'s effort to provide a definition for subliminal techniques [2], Franklin et al.'s proposed definitions for subliminal, purposefully manipulative, and deceptive techniques [3], Bulgakova's analysis of the prohibition on the use of subliminal techniques [4], and Leiser's comparative analysis of prohibited uses that deploy manipulative techniques in different mandates of the Act [12]. However, the recent publication of the Commission's guidelines on prohibited AI systems [5] has addressed several issues previously highlighted in these studies. Since the official publication of the AI Act and the Commission guidelines on prohibited AI, there are only few studies published including Barkane and Buka's critical analysis of the prohibitions of surveillance and predictive policing [13]. In general, the body of work on the criteria for prohibited systems is mainly focused on clarification of the wording of the Act's text and none of the aforementioned studies, in addition to the Commission's guidelines, establish a *holistic view* of the prohibited categories, nor do they identify the set of concepts of AI use cases that make them prohibited.

In regard to the **codification** of rules for AI Act's risk categorisation, the *Decision-Tree-based framework* [14] is a static framework that aims to assist in classification of AI systems based on the AI Act. The framework is based on a decision tree comprising 20 questions for determining the risk category associated with an AI system. Our pervious work [10] identifies 5 concepts to facilitate identification of high-risk AI systems according to Annex III, which are: domain, purpose, AI capability, deployer, AI subject. We further codified the rules using SHACL to enable automated determination of such systems. Given the interest our work on high-risk AI has attracted, in this paper we follow the same approach for prohibited practices.

## 3. Methodology

Identification of classification rules for the AI Act's prohibited AI practices is guided by our contributions in [10]. In this previous work, through manual annotation of Annex III of the AI Act, we identified the minimum set of information elements (the 5 aforementioned concepts) required to determine high-risk applications of AI. Building upon these identified information elements, we take the following steps to create a framework for determining prohibited AI practices (see section 4):

1. Identify the 5 concepts (domain, purpose, AI capability, deployer, AI subject) from each prohibited condition described in Art. 5(1),
2. Determine whether the 5 concepts are sufficient to describe prohibited AI practices in a unique way that sufficiently distinguish them from each other,
3. Where the 5 concepts are not sufficient, identify the minimal set of additional concepts needed for describing the prohibited AI condition.

To be able to provide open data specifications for prohibited systems, we add the identified additional concepts (step 3) to the AI Risk Ontology (AIRO) [15][1] and further populate the Vocabulary of AI Risks (VAIR)[2] with the instances identified from the annotation process.

Demonstrating how the prohibited AI rule-checking can be automated for supporting compliance tasks, we utilise existing Semantic Web languages and standards with rule-checking capabilities. While there are multiple languages and standards offering such capabilities, including the Shapes Constraint Language (SHACL) [8], the Semantic Web Rule Language (SWRL) [16], N3 (Notation3) rules [9], and the Shape Expressions (ShEx) language [17], we use SHACL in this work as it is a W3C recommended language. We also use N3 to express rules in a simplified if-then style manner to address the complexity of expressing the rules using SHACL (see section 5).

## 4. Patterns of Prohibited AI Practices under the AI Act

The analysis Art. 5(1) aims to identify the minimum set of concepts that are adequate to uniquely describe prohibited AI practices. Following the steps outlined above, Art. 5(1) clauses were manually annotated to identify the 5 following concepts: domain, purpose, AI capability, deployer, AI subject. Then, additional concepts were identified in each clause. An example of annotating Art. 5(1a) is shown in Figure 1. The manual annotation was carried out by the lead author and validated through discussions with co-authors.

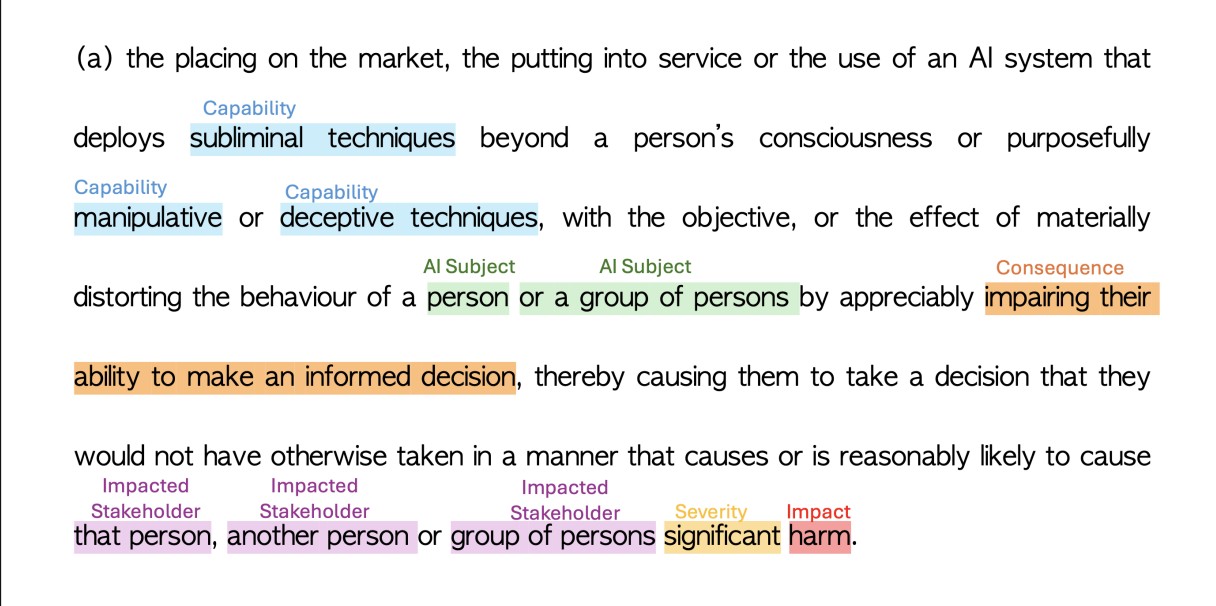

**Figure 1:** Annotation of prohibited AI practice described in Art. 5(1a)

The annotation exercise revealed that among the 5 previously identified concepts, AI deployer is not a decisive factor in determining prohibited AI systems. Additionally, we identified the following additional concepts: *data processed by the system*, *locality of use*, *consequence*, *impact and its severity*, and *impacted stakeholder(s)*. *Locality of use* defines the environment in which the system is used, e.g. work place. *Consequence* refers to the direct immediate effect of using an AI system, whether it leads to harms to individual, groups, and society or not. *Impact* refers to the overall ultimate effect of an AI system on *impacted stakeholders*, such as individual, groups, and society. We treat the combination of consequence, impact and its severity, and impacted stakeholder as *(harmful) risk requirement*

---

[1]https://w3id.org/airo
[2]https://w3id.org/vair

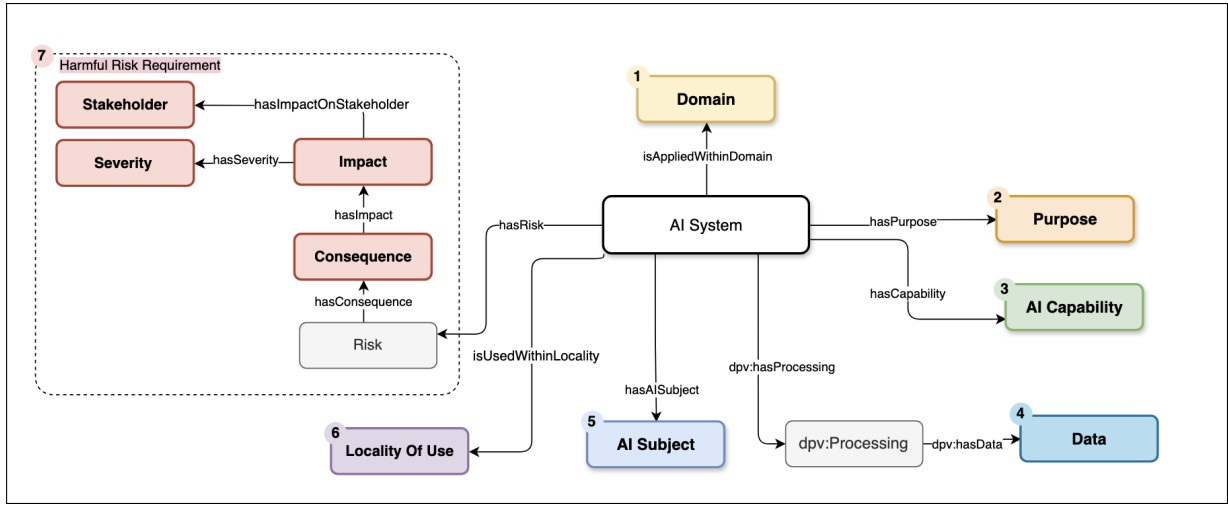

**Figure 2:** Semantic model of concepts (from AIRO) required for determining prohibited AI systems as per Art. 5(1)

on the basis that these concepts can only be determined through risk assessment. Our analysis shows that among the prohibited conditions in Art. 5(1), points (a), (b), and (c) depend on the results of a risk assessment process that identifies consequences, associated impacts, their severity, and the stakeholders affected.

The minimal set concepts for determining prohibited AI systems are expressed in a form of questions in the following:

1. In which **domain** is the AI system used?
2. What is the **purpose** of using the AI system?
3. What is the **capability** of the AI system?
4. What **data** is processed by the AI system?
5. Who is the **AI subject**?
6. What is the **locality of use**?
7. what is the **harmful risk** caused by the AI system?
    a) What is the **consequence** of using the system?
    b) What is the **impact** of using the AI system?
    c) What is the **severity of the impact**?
    d) Who is the **impacted stakeholder**?

These concepts and their relations are modelled in our previously developed ontology for AI risks, AIRO, and are illustrated in Figure 2. As shown in the figure, concepts from the Data Privacy Vocabulary (DPV) [18] are reused for expressing the data processed by the system.

The detailed analysis of the prohibited conditions is presented in Appendix A and a summary of the conditions is illustrated in Figure 3. It should be noted that in our analysis of Art. 5(1) points (a) and (b), we consider *materially distorting behaviour* as a consequence rather that purpose of the system, even though the wording of the AI Act suggests that it can be either an *objective* or an *effect* of employing the AI system. This interpretation is based on the reality that AI providers rarely, if ever, explicitly state that their system's purpose is "behaviour distortion" or "impairing decision making". Further, in development of emerging technologies such effects of AI are often identified after deployment as (unintended) consequences.

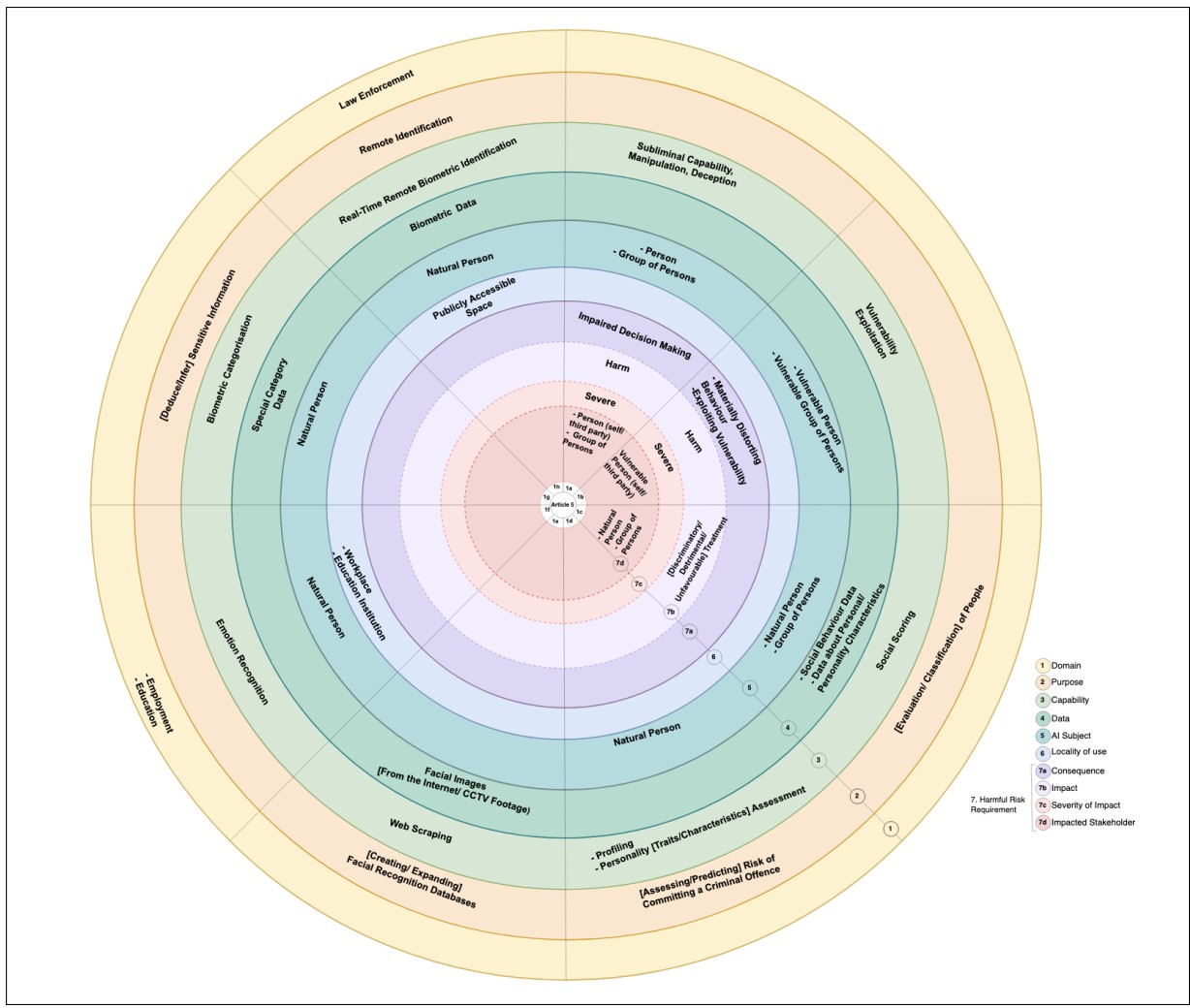

**Figure 3:** Patterns of prohibited AI practices

## 5. Codified Rules for Determining Prohibited AI Practices

In our framework, prior to rule-checking, an RDF-based specification of an AI systems should be created to enable determination of its risk category. In Listing 1, machine-readable specification of an AI chatbot that impersonates a friend of a person for scamming, described in the Commission's guideline [5], is shown[3]. This specification serves as a *data graph* that can be validated against both *shape graphs*, which describe the rules using SHACL for prohibited AI systems, and *N3 rules*.

To show how SHACL can be used for describing prohibited rules, we provide an example of a shape graph specifying Art. 5(1a) condition in Listing 2. As it is clear in the listing, the shape graph is expressed as negation of the condition using sh:not. This is due to the fact that SHACL's validation report (sh:ValidationResult) is only generated in case of non-conformance. We used the validation report to enhance transparency by providing guiding information about the clause based on the which the system is determined to be prohibited. The SHACL shapes for prohibited AI systems are published on GitHub[4] under permissive licences.

---

[3]This use case, along with additional examples, is available at: https://github.com/DelaramGlp/airo/tree/main/usecase
[4]https://github.com/DelaramGlp/airo/tree/main/prohibited-shacl

```
1   @prefix rdf: <http://www.w3.org/1999/02/22-rdf-syntax-ns#> .
2   @prefix airo: <https://w3id.org/airo#> .
3   @prefix vair: <https://w3id.org/vair#> .
4   @prefix dpv: <https://w3id.org/dpv#> .
5   @prefix ex: <https://example.com/> .
6   @prefix risk: <https://w3id.org/dpv/risk#>.
7
8   ex:ai_chatbot a airo:AISystem ;
9       airo:hasPurpose ex:engage_in_human_like_conversation ;
10      airo:hasCapability ex:impersonation ;
11      airo:hasAISubject ex:chatbot_user ;
12      airo:hasRisk ex:risk_of_fraud;
13      dpv:hasProcessing ex:processing_conversation .
14
15  ex:engage_human_like_conversation a airo:Purpose .
16
17  ex:impersonation a airo:Capability , vair:DeceptiveTechnique .
18
19  ex:chatbot_user a airo:AISubject, dpv:DataSubject, vair:NaturalPerson .
20
21  ex:processing_conversation a dpv:Processing ;
22      dpv:hasData ex:voice .
23
24  ex:voice a dpv:PersonalData .
25
26  ex:risk_of_fraud a airo:Risk ;
27      airo:hasConsequence ex:victim_tricked_into_transfering_money .
28
29  ex:victim_tricked_into_transfering_money a airo:Consequence, vair:ImpairedDecisionMaking ;
30      airo:hasImpact ex:financial_loss .
31
32  ex:financial_loss a airo:Impact, vair:Harm ;
33      airo:hasSeverity risk:ExtremelyHighSeverity ;
34      airo:hasImpactOnStakeholder ex:chatbot_user .
35
```

Listing 1: RDF-based specification of the AI chatbot example

As shown in the listing expressing the the harm requirement within a SHACL shape graph requires nested `NodeShapes` which adds to complexity of the shape and further effects its readability and performance. To address this issue, we use N3 to provide more flexible and simplified representation of the rules. Listing 3 illustrates the encoding of the Art. 5(1a) in N3. For simplicity, the listing is restricted to `NaturalPersons` as AI subjects and impacted stakeholders. The N3 rules are made available online[5].

---

[5]https://github.com/DelaramGlp/airo/tree/main/prohibited-n3

```
1   @prefix rdf: <http://www.w3.org/1999/02/22-rdf-syntax-ns#> .
2   @prefix rdfs: <http://www.w3.org/2000/01/rdf-schema#> .
3   @prefix sh: <http://www.w3.org/ns/shacl#> .
4   @prefix airo: <https://w3id.org/airo#> .
5   @prefix vair: <https://w3id.org/vair#> .
6   @prefix terms: <http://purl.org/dc/terms/> .
7   @prefix ex: <https://example.com/ns#> .
8   @prefix risk: <https://w3id.org/dpv/risk#>.
9   ex:Art5-1-a
10      a sh:NodeShape ;
11      sh:targetClass airo:AISystem ;
12      sh:message "Prohibited as per AI Act, Art. 5(1a): AI system that deploys subliminal techniques
        ↪   beyond a person's consciousness or purposefully manipulative or deceptive techniques, with
        ↪   the objective, or the effect of materially distorting the behaviour of a person or a group
        ↪   of persons by appreciably impairing their ability to make an informed decision, thereby
        ↪   causing them to take a decision that they would not have otherwise taken in a manner that
        ↪   causes or is reasonably likely to cause that person, another person or group of persons
        ↪   significant harm"@en ;
13      sh:description "AI systems that AI system that deploys subliminal techniques beyond a person's
        ↪   consciousness or purposefully manipulative or deceptive techniques"@en ;
14      sh:not [
15          sh:and (
16                  sh:property [
17                      a sh:PropertyShape ;
18                      sh:path airo:hasCapability ;
19                      sh:class vair:DeceptiveTechnique ; ]
20                  sh:property [
21                      a sh:PropertyShape ;
22                      sh:path airo:hasAISubject ;
23                      sh:or (
24                          [ sh:class vair:NaturalPerson ; ]
25                          [ sh:class vair:Group; ] )]
26                  sh:property [
27                      a sh:PropertyShape ;
28                      sh:path airo:hasRisk ;
29                      sh:node [
30                          a sh:NodeShape ;
31                          sh:targetClass airo:Risk ;
32                          sh:property [
33                              sh:path  airo:hasConsequence ;
34                              sh:class vair:ImpairedDecisionMaking ;
35                              sh:node [
36                                  a sh:NodeShape ;
37                                  sh:targetClass airo:Consequence ;
38                                  sh:property [
39                                      sh:path airo:hasImpact ;
40                                      sh:class vair:Harm ;
41                                      sh:node [
42                                          a sh:NodeShape ;
43                                          sh:targetClass airo:Impact ;
44                                          sh:property [
45                                              sh:path  airo:hasSeverity ;
46                                              sh:hasValue  risk:ExtremelyHighSeverity ;]  ;
47                                          sh:property [
48                                              sh:path airo:hasImpactOnStakeholder;
49                                              sh:class vair:NaturalPerson ;
50                                              #For brevity, vair:Group is omitted
51                                              ] ] ]]]])].
52
```

Listing 2: SHACL shape for identifying prohibited AI systems from Art. 5(1a)

```
1   @prefix airo: <https://w3id.org/airo#> .
2   @prefix vair: <https://w3id.org/vair#> .
3   @prefix risk: <https://w3id.org/dpv/risk#>.
4   @prefix ex: <https://example.com/ns#> .
5
6   {
7       ?system airo:hasCapability ?capability .
8       ?capability a vair:DeceptiveTechnique.
9       ?system airo:hasAISubject ?subject .
10      ?subject a  vair:NaturalPerson .
11      ?system airo:hasRisk ?risk .
12      ?risk airo:hasConsequence ?consequence .
13      ?consequence a vair:ImpairedDecisionMaking .
14      ?consequence airo:hasImpact ?impact .
15      ?impact a vair:Harm .
16      ?impact airo:hasSeverity risk:ExtremelyHighSeverity .
17      ?impact airo:hasImpactOnStakeholder ?stakeholder .
18      ?stakeholder a vair:NaturalPerson .
19
20  } => { ?system a ex:prohibited-5-1a . } .
21  .
22
```

Listing 3: N3 rule for identifying prohibited AI systems as per Art. 5(1a)

## 6. Limitations

As mentioned earlier, an initial validation of the analysis of prohibited practices, i.e. results of the manual annotation, was conducted. However, further consultation with subject matter experts, including lawyers and policymakers, is required to ensure the validity of our interpretation of the AI Act. Nevertheless, since our proposed framework for determining prohibited practices leverages Semantic Web technologies, it is flexible and can accommodate future enhancements.

In the case of our research, manual annotation of clauses describing prohibited practices was possible given the limited number of these clauses. However, manually annotating a large number of AI use cases to determine their risk level under the AI Act might not be possible. To address this challenge, a combination of Large Language Models (LLMs) and ontologies can provide a scalable solution. However, this requires appropriate measures to avoid hallucinations.

Our proposed framework is designed to support regulatory simplification and automation by adopting an open, standards-based, and interoperable approach. It is important to not that our framework does not substitute legal advice and determining some of the concepts, in particular the risk requirement, require legal interpretation as well as technical analysis. Given the high stakes involved in determining risk levels under the AI Act, our framework should be viewed as a supporting tool to assist in identifying prohibited practices, not as a replacement for legal expertise.

## 7. Conclusion and Future Work

In this paper, we presented a Semantic Web-based framework to assist with determining prohibited AI systems according to the AI Act. This paper followed the approach we took in our previous work for determining high-risk applications [10] in terms of both conceptualisation and codification. Although these two studies are aligned and complementary, they have not yet integrated to capture the interplay between the two categories. Thus, in our future work, we aim to address this gap by incorporating the exceptions to prohibited systems, given that these exceptions are mostly result in the system being classified as high-risk [5]. For those AI systems listed in Annex III (high-risk AI systems) but may also meet the prohibited conditions, and therefore be classified as prohibited, a sequential classification

wherein determining prohibited AI supersedes high-risk AI may be appropriate.

In our future work, we also aim to include the specificities from the Commission's guidelines on prohibited systems [5] and further populate VAIR, for example with instances of subliminal techniques, including visual subliminal messages, subvisual and subaudible cueing, and misdirections. We also plan to propose these concepts for inclusion within DPV.

## Acknowledgments

This work has received funding from the European Commission's Horizon Europe Research and Innovation Programme under grant agreement No. 101177579 (FORSEE), the European Union's Horizon 2020 research and innovation programme under the Marie Skłodowska-Curie grant agreement No. 813497 (PROTECT ITN), and from the ADAPT Centre for Digital Media Technology, which is funded by Research Ireland and is co-funded under the European Regional Development Fund (ERDF) through Grant#13/RC/2106_P2. Harshvardhan J. Pandit is a member of AI Accountability Lab, which is funded under John D. and Catherine T. MacArthur Foundation grant with project #216001 and award #19034.

## Declaration on The Use of Generative AI

During the preparation of this work, the first author used OpenAI's ChatGPT and Anthropic's Claude for language refinement and Microsoft's Copilot for code debugging assistance. These tools were used in a limited capacity and lead author reviewed and edited the generated content as needed and takes full responsibility for the publication's content.

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

# A. Detailed Analysis of Prohibited AI Practices

**Table 1**

Analysis of prohibited AI practices listed in Art. 5, Points (1a) to (1e)

| Art. 5 clause | Concepts |
|---|---|
| (1a) | 1. **Domain**: Any
2. **Purpose**: Any
3. **Capability**: *Subliminal Capability, Manipulation, Deception*
4. **Data processed**: Any
5. **AI subject**: *Natural Person, Group of Persons*
6. **Locality of use**: Any
7a. **Consequence**: *Impaired Decision Making*
7b. **Impact**: *Harm*
7c. **Severity of impact**: *Severe*
7d. **Impacted stakeholder**: *Natural Person* (self or third-party), *Group of Persons* |
| (1b) | 1. **Domain**: Any
2. **Purpose**: Any
3. **Capability**: *Exploitation Of Vulnerability*
4. **Data processed**: Any
5. **AI subject**: *Vulnerable Person, Vulnerable Groups Of Persons*
6. **Locality of use**: Any
7a. **Consequence**: *Materially Distorting Behaviour, Exploiting Vulnerability*
7b. **Impact**: *Harm*
7c. **Severity of impact**: *Severe*
7d. **Impacted stakeholder**: *Vulnerable Person* (self or third-party) |
| (1c) | 1. **Domain**: Any
2. **Purpose**: *Evaluation Of People, Classification Of People*
3. **Capability**: *Social Scoring*
4. **Data processed**: *Social Behaviour Data,Known, Inferred or Predicted Personal Characteristics, Known, Inferred or Predicted Personality Characteristics*
5. **AI subject**: *Natural Person, Group of Persons*
6. **Locality of use**: Any
7a. **Consequence**: Any
7b. **Impact**: *Discriminatory Treatment, Detrimental Treatment, Unfavourable Treatment*
7c. **Severity of impact**: Any
7d. **Impacted stakeholder**: *Natural Person, Group of Persons* |
| (1d) | 1. **Domain**: Any,
2. **Purpose**: *Assessing Risk of Committing a Criminal Offence, Predicting Risk of Committing a Criminal Offence*
3. **Capability**: *Profiling, Personality Trait Analysis, Personality Characteristics Assessment*
4. **Data processed**: Any
5. **AI subject**: *Natural Person*
6. **Locality of use**: Any
7a. **Consequence**: Any
7b. **Impact**: Any
7c. **Severity of impact**: Any
7d. **Impacted stakeholder**: Any |
| (1e) | 1. **Domain**: Any
2. **Purpose**: *Creating Facial Recognition Databases, Expanding Facial Recognition Databases*
3. **Capability**: *Web Scraping*
4. **Data processed**: *Facial Images From The Internet, Facial Images From CCTV Footage*
5. **AI subject**: *Natural Person*
6. **Locality of use**: Any
7a. **Consequence**: Any
7b. **Impact**: Any
7c. **Severity of impact**: Any
7d. **Impacted stakeholder**: Any |

**Table 2**

Analysis of prohibited AI practices listed in Art. 5, Points (1f) to (1h)

| Art. 5 clause | Concepts |
|---|---|
| (1f) | 1. **Domain**: *Employment, Education*
2. **Purpose**: Any
3. **Capability**: *Emotion Recognition*
4. **Data processed**: Any
5. **AI subject**: *Natural Person*
6. **Locality of use**: *Workplace, Education Institution*
7a. **Consequence**: Any
7b. **Impact**: Any
7c. **Severity of impact**: Any
7d. **Impacted stakeholder**: Any |
| (1g) | 1. **Domain**: Any
2. **Purpose**: *Deduce Sensitive Information, Infer Sensitive Information*
3. **Capability**: *Biometric Categorisation*
4. **Data processed**: *Special Category Data*
5. **AI subject**: *Natural Person*
6. **Locality of use**: Any
7a. **Consequence**: Any
7b. **Impact**: Any
7c. **Severity of impact**: Any
7d. **Impacted stakeholder**: Any |
| (1h) | 1. **Domain**: *Law Enforcement*
2. **Purpose**: *Remote Identification*
3. **Capability**: *Real-Time Remote Biometric Identification*
4. **Data processed**: *Biometric Data*
5. **AI subject**: *Natural Person*
6. **Locality of use**: *Publicly Accessible Space*
**Consequence**: Any
7a. **Consequence**: Any
7b. **Impact**: Any
7c. **Severity of impact**: Any
7d. **Impacted stakeholder**: Any |