# OpenReview forum: "Semantic Patterns of Prohibited AI Systems in the EU AI Act"
_SEMANTiCS.cc/2025/Workshop/NXDG — NXDG 2025_

### Official Review · ~Anelia_Kurteva1 · 2025-07-18
**Deriving semantic patterns for AI systems that are considered prohibited according to the EU AI Act**

**Rating:** 7
**Confidence:** 4

**Review:**

In this paper, the authors derive semantic patterns for AI systems that are considered prohibited according to the EU AI Act. I found the paper easy to follow and enjoyed reading it.

Comments:
Was the annotation of the prohibited AI practices shown on Figure 1 manual, semi-automated or fully automated? If manual, then this should be noted as a limitation due to the scalability of the approach. The authors can also discuss how generative AI can help automate annotations.

Unclear if the patterns are openly accessible and documented somewhere online to support reuse.

I am missing discussion on the evaluation and validation of the work. The motivation could be better reinforced in the paper as well. Who will benefit from this seemingly valuable work?

Future work and some limitations have been mentioned. A discussion on the implementation of the approach could be useful.

Minor comments:
There are some issues with capitalisation in the text and the references (2, 11, 15).
Missing "the" in front of "official AI journal" at the start of section 2.
Figure 3 is unreadable when printed on paper.

---

### Official Review · ~Jakob_Merane1 · 2025-07-22
**Overall, this is a strong contribution that is well enough conducted and will be clearly of interest to the participants of the workshop. The paper is clear, systematic, and original. To further strengthen the work, I recommend clarifying the annotation methodology and being more explicit about the limitations regarding legal interpretation and automation.**

**Rating:** 8
**Confidence:** 3

**Review:**

First of all, I want to applaud the authors for addressing this timely topic. They do so in a very clear and systematic way with a well-structured and convincing manuscripts. I really enjoyed reading it (and your previous research)!

With respect to the criteria of originality and significance, the manuscript is already in good shape. However, here are some thoughts on how to further improve the quality and clarity.

- I am mostly concerned about the methodology. This paper would benefit from more details, both in the manuscript and Appendix, on how the annotation process and the identification of the concepts was done. I'd suggest (if possible) bringing it more in accordance with social science standard. For example, it could be good to have two annotators for a subset of articles and calculate inter-annotator agreement. If that's not possible, I'd at least consider making the annotations available. Overall, it would be helpful to describe the process and all the design choices in more detailed (for instance: why you decided to keep the additional concepts to a "minimal set"? etc.). As a trained lawyer, I'm not sure whether I'd prefer a minimal set over a comprehensive set.

- I also think that, at times, the paper slightly overstates its contribution. For example, the claim of "open, interpretable, auditable, automated implementation and enforcement of AI Act" may not fully account for the remaining need for legal interpretation of certain terms. Having experience at the intersection of CS (including formal methods) and law, I do think it would be wise to make the implicit disclaimers explicit, which means that certain rules and terms (significant harm, impairing decision-making etc.) still depend on legal interpretation. I strongly recommend adding a limitation section/sentence which makes clear which problems you solve and the boundaries.

Here are some minor comments/typos:
-  Art. 50 .
- to facilitate identification (of) prohibited
- , (that) have been

---

### Decision · Program_Chairs · 2025-07-25

Accept